# Comparison of the Electrochemical Response of Carbon-Fiber-Reinforced Plastic (CFRP), Glassy Carbon, and Highly Ordered Pyrolytic Graphite (HOPG) in Near-Neutral Aqueous Chloride Media

**Stanley Udochukwu Ofoegbu [1,2,*], Mário Guerreiro Silva Ferreira [2], Helena I. S. Nogueira [3] and Mikhail Zheludkevich [4,5]**

[1] Centre for Mechanical Technology and Automation (TEMA), Department of Mechanical Engineering, University of Aveiro, Campus Universitário de Santiago, 3810-193 Aveiro, Portugal
[2] Department of Materials and Ceramic Engineering, CICECO-Aveiro Institute of Materials, University of Aveiro, Campus Universitário de Santiago, 3810-193 Aveiro, Portugal
[3] Department of Chemistry, CICECO-Aveiro Institute of Materials, University of Aveiro, Campus Universitário de Santiago, 3810-193 Aveiro, Portugal
[4] Institute of Surface Science, Helmholtz-Zentrum Hereon, Max-Planck-Strasse 1, 21502 Geesthacht, Germany
[5] Institute for Materials Science, Faculty of Engineering, Kiel University, 24103 Kiel, Germany
**\*** Correspondence: ofoegbu.stanley@ua.pt; Tel.: +351-915-248-642

**Abstract:** Carbon-fiber-reinforced polymers (CFRP), being conductive, are capable of supporting cathodic oxygen reduction reactions (ORR) and thus promote galvanic corrosion when coupled to many metallic materials. Hence, understanding cathodic processes at carbon surfaces is critical to developing new strategies for the corrosion protection of multi-material assemblies. In the present work, the electrochemical responses of CFRP, glassy carbon, and HOPG (Highly Ordered Pyrolytic Graphite) have been evaluated in a quiescent 50 mM NaCl solution, and their respective activities towards ORR have been ranked. Employing the averages of the specific charges (CFRP, 129.52 mC cm$^{-2}$; glassy carbon, 89.95 mC cm$^{-2}$; HOPG, 60.77 mC cm$^{-2}$) passed during 1 h polarization of each of the 3 carbon surfaces at −1000 mV$_{SCE}$ in the test media as a ranking criterion, the propensities of the 3 carbon surfaces (CFRP, GC, and HOPG) to support cathodic activities that can lead to anodic metal dissolution on galvanic coupling to metallic materials are ranked thusly; CFRP > GC > HOPG. This ranking is consistent with the trend of capacitance values obtained in this work: CFRP (19.5 to 34.5 µF cm$^{-2}$), glassy carbon (13.6 to 85.5 µF cm$^{-2}$), and HOPG (1.4 to 1.8 µF cm$^{-2}$). A comparison of electrochemical data at potentials relevant to galvanic coupling to metals indicated that at these cathodic potential(s) the CFRP surface is the most electrochemically active of the studied carbon surfaces. On the basis of the values and trends of the electrochemical parameters evaluated, it is postulated that the observed differences in the electrochemical responses of these 3 carbon-rich surfaces to ORR are significantly due to differences in the proportions of edge sites present on each carbon surface. These results could provide valuable insights on plausible strategies for designing carbon surfaces and carbon fiber composites with reduced activity toward ORR for corrosion protection applications or enhanced activity towards ORR for energy applications.

**Keywords:** galvanic coupling; CFRP; carbon surfaces; cathodic polarization; oxygen reduction reactions (ORR); multi-material combinations

## 1. Introduction

Carbon, and particularly carbon surfaces, are in high demand for a wide range of technological applications. Carbon materials are used in a variety of applications as anode

or cathode materials due to their low cost and chemical stability in a variety of environments, good thermal and electrical conductivity, and wide potential window (2.5 to 3.5 V [1,2]). Carbon electrodes and surfaces are employed in a wide variety of applications including structural reinforcements [3–5], electroanalysis (i.e., sensor applications) [6–9], electrosynthesis [10–12], energy applications [13,14], environmental remediation and water treatment [13,15], and many more. The use of carbon in the form of carbon fibers in electrochemical systems can be considered a miniaturization step that yields a variety of advantages: the possibility of concurrent electroanalysis in small volumes and at high speeds with enhanced signal quality due to the "microelectrode effect" [16,17] and the subsequent emergence of micro-electrode arrays (MEAs) [18–20]. The incorporation of nanostructured carbon such as nanotubes into carbon-reinforced composites [21,22], the growth of carbon nanotubes on carbon fiber [23,24], and other carbon substrate surfaces [25,26] have been reported to yield remarkable improvements in electrochemical activity [24], and subsequently led to the emergence of carbon-based nanoelectrode arrays with a very wide range of applications [26,27].

In a lot of these applications, the chemical and electrochemical reactivity of carbon surfaces is an important performance-determining indicator. Hence, insights into the electrochemical response of a variety of carbon surfaces are important in order to maximize material surface capabilities in the design of highly efficient application-specific engineered carbon surfaces. In this work, the electrochemical response of three carbon-rich surfaces (glassy carbon (GC), highly ordered pyrolytic graphite, and carbon fiber-reinforced polymer composite surface (CFRP)) towards oxygen reduction reactions (ORR) in quiescent aqueous media in configurations consistent with conditions in many corrosion systems has been studied, and the respective responses have been compared. Carbon-fiber-reinforced polymer composites are often components of multi-material structures increasingly employed in the aeronautical and automobile industries in a bid to realize the weight-to-strength ratios necessary to achieve reduced emission targets. In such multi-material combinations, the galvanic coupling of conductive carbon surfaces to metallic materials is likely [28]. In this configuration, galvanic corrosion (anodic dissolution) of the coupled metal is promoted by cathodic reactions (primarily ORR) that take place on conductive carbon surfaces that act as a cathode.

Glassy carbon has been described as a graphene-rich form of elemental carbon derived from carbon-rich polymer pyrolysis and comprising three-dimensionally arranged curved graphene fragments together with fractions of disordered carbon and voids [29,30]. The presence of voids in glassy carbon is thought to account for its low density ($\approx$1.5 g cm$^{-3}$) in comparison to other graphites ($\approx$2.3 g cm$^{-3}$) [29,31]. Glassy carbon is regarded as a chemically stable form of entirely sp$^2$-bonded carbon with locally ordered domains and is considered to be the intermediate material between graphite and diamond [32]. Glassy carbon is reported to contain graphite-like nanostructures that, depending on its mode of production, can present as a complex solid containing mixtures of micrographitized and fullerenized zones where sp$^2$ carbon atoms predominate [33–36].

Pyrolytic graphites manifest a high degree of anisotropy in both their electrical and heat conductivity and are impervious to gases [37,38]. Pyrolytic graphites are characterized by large strains in the c directions and a preferred orientation that increases with increasing deposition temperature (e.g., <102:1 to as high as 104:1 as the temperature is increased from 1700 °C to 2500 °C), nearly random layer order (turbostratic), average crystallite dimensions (Lc) ranging from below 100 Å to up to 265 Å depending on deposition temperature, and a relatively strain-free basal plane [38]. Highly oriented pyrolytic graphite (HOPG) is a synthetic graphite prepared by the thermal and/or stress annealing of pyrolytic graphite [39]. The thermal and/or stress annealing of pyrolytic graphite at high temperatures ($\geq$ 2500 °C) results in a material (HOPG) with more uniform interlayer spacing ($\approx$ 3.35 Å) across its thickness and a drastically reduced stacking disorder factor/mosaic spread (the angle between the tiles of graphite) that tends towards zero) [40]. Treated pyrolytic graphite with a mosaic spread of less than 1 is classified as HOPG [41]. HOPG is

often used as a well-defined model carbon material for studying the chemical and electrochemical behavior of carbon surfaces [42,43].

Oxygen reduction on carbon surfaces is of great relevance in a variety of applications. In some applications, like fuel cells and other energy applications, a high rate of ORR on carbon surfaces is desirable. However, in other applications, like the corrosion of metals galvanically coupled to metallic materials (multi-material combinations), it is undesirable and needs to be prevented. Whether desirable or undesirable, a better understanding of the mechanism(s) and kinetics of ORR on carbonaceous surfaces under application-relevant conditions is critical for manipulating the ORR phenomenon on carbon surfaces to advantage. Consequently, application-specific designs of carbon surfaces require information-driven surface engineering or modification(s). Due to their potential benefits, there has been sustained research interest in metal-free carbon-based electrocatalysts for oxygen reduction reactions [44–49] as replacements for costly Pt-based ORR catalysts. The advantages of carbon over other electrocatalysts include its stability in neutral solution, low cost, abundance, good electrical conductivity, and light weight [50], making it a good candidate material for the oxygen reduction reactions required in fuel cells.

Carbon-fiber-reinforced polymers (CFRPs) are composite materials made of carbon fibers in a matrix of polymers, usually, epoxy, polyester, and vinyl ester. The carbon fibers can be continuous or discontinuous and can be arranged in a variety of orientations [51,52]. Due to the high strength-to-weight ratios achievable, CFRPs are replacing metallic materials in a lot of applications, particularly in the aeronautical and automobile industies. Carbon-fiber-reinforced polymer (CFRP) is much exploited in the aeronautical [53–56] and automobile [57–62] industries in weight-optimized multi-material combinations that offer high strength-to-weight ratios. In such CFRP-containing multi-material combinations, galvanic coupling of the metal or alloy(s) component(s) with the conductive CFRP composite is often inevitable. Such galvanic coupling of CFRP to the metallic component(s) of multi-material assemblies results in accelerated anodic metal dissolution due to the ability of CFRP (actually, its carbon surfaces) to support cathodic reactions such as oxygen reduction.

Irrespective of the history of any carbon material, carbon surfaces are known to present two types of surface sites with different reactivities: edge-plane sites and basal plane sites [63,64]. Due to the known differences in the electrochemical behavior of edge and basal sites [63–65], their technological applications vary. In all applications, the carbon surface presents either predominantly edge sites, basal sites, or a combination of both. It is on the more electrochemically active edge-plane sites [63,66] that most electrochemical activity (including ORR) predominantly occurs on carbon and carbon-rich surfaces.

Chu and Kinoshita [63] have demonstrated that the electrochemical behavior of carbon surfaces can be enhanced by surface modifications that introduce more edge sites or defects on the carbon surface(s). In contrast, it has been demonstrated that the electrochemical behavior of carbon-rich surfaces can be suppressed by surface modifications that employ surfactants that tend to interact and block these active sites [67–72] or by precipitating inhibitors that deposit on carbon surfaces and thus interfere with the transport of electroactive species to the active sites [73–75].

ORR involves electron transfer steps. It is reported [76] that a carbon matrix with a low degree of graphitization is not conducive to electron transfer and conduction. Some reports [77–79] indicate that the electrochemical oxidation of carbon (glassy carbon) activates the electrode surface, leading to faster electron transfer kinetics, which is desirable for electrochemical sensor applications. Interestingly, Matsumoto et al. [80] prepared contaminant-free graphene oxide (eGO) electrodes via controlled oxidation (by electrolysis in pure water under high voltage) and electrochemically reduced the graphene oxide (eGO) electrodes to obtain reduced eGO (r-eGO). The reduced graphene oxide (r-eGO) thus obtained is reported to display high double-layer charging capacitance, high electrocatalytic activity for oxygen reduction reactions (ORR), and n-type semiconductor electrode behavior. They attributed [80] these excellent properties of reduced graphene oxide (r-eGO) to

carbon defects and/or OH groups produced on the reduction of epoxide groups formed at the basal plane. Carbon surfaces are sought after as metal-free electrocatalysts of oxygen reduction reactions (ORR) in alkaline media [81–84], and reports suggest that surface modification to achieve the presence of nitrogen-containing surface groups on carbon materials enhances catalytic activity towards ORR [85–89].

Under sufficient applied polarization and/or in the presence of strong oxidizers [90–93], the graphite layers on carbon surfaces can be oxidized. This oxidation results in oxidized graphitic sheets with the significant presence of hydroxyl (-OH), ketone (C-O), carbonyl (C=O), carboxylic (O=C-O), and epoxide groups on their graphite surface [93–99] and highly oxidized polycyclic molecules (tannic acids or oxidized debris) that apparently result from side reactions during graphite oxidation [93,100]. Through $\pi-\pi$ interactions and hydrogen bonding [93], this debris can remain adsorbed to the oxidized graphite or graphene surface and thus influence the physical and electrochemical responses (electron transfer kinetics) of these carbon surfaces. Wu et al. [101] carried out the nitric acid oxidation of carbon fibers and reported the formation of acidic functional groups on PAN-based high-strength carbon fibers and the adherence of partially oxidized graphitic fragments that are insoluble in water. However, they reported that these partially oxidized graphitic fragments are sufficiently solvated and dissolved in aqueous sodium hydroxide and hence are removable by contact with NaOH (high pH).

Literature data suggest that the formation of these surface species is the initial step in the degradation of carbon surfaces. Yi et al. [102] investigated the electrochemical corrosion of a glassy carbon electrode in alkaline, neutral, and acidic media under conditions similar to those which occur during oxygen reduction reactions (ORR) and oxygen evolution reactions (OER) and discovered that glassy carbon degradation begins with the formation of surface oxides via acid-catalyzed process(es) in acidic media. This process(es) leads to ring opening in the graphitic structure and ultimately oxidation in the bulk material. With respect to degradation in alkaline media, they [102] posited that this occurs through OH radicals' preferential attack on alkyl side chains, which leads to the oxidation of the edges of carbon layers; hence, they become hydrophilic and dissolve.

Literature on the cathodic modification of carbon surfaces appears to be scarce. However, insights from previous works on carbon surface polarization in aqueous media indicate that, under cathodic polarization, the local environment around carbon surface(s) is amenable to being different from the bulk environment and becoming alkaline. Furthermore, [28,51,67] guides us to an appreciation of feasible changes to carbon surfaces subjected to cathodic polarization. A report [80] indicates that, at the limits of the cathodic potentials employed in this work, the electrochemical reduction of any graphitic oxides formed by prior anodic polarization or oxidation might be feasible. The alkaline oxidation of carbon surface(s) is deemed to operate through mechanism(s) that are different from its oxidation in acidic media. For oxidation in acidic media, Yi et al. [103] advanced an acid-catalyzed electrophilic substitution reaction mechanism. Other authors proposed a variety of mechanisms for carbon fiber (carbon surface) oxidation in an alkaline environment, based primarily on OH radicals promoting oxidation [103–105], and oxygen (as an oxidizing agent) enabling the oxidation of the carbon fiber anode in the case of water electrolysis [104,106–108].

The major cathodic process on electrodes immersed in aqueous media under ambient conditions relevant to galvanic coupling with metals is the oxygen reduction reaction (ORR). Based on the electrode material and the solution composition (and pH), the oxygen reduction reaction proceeds either as a direct four-electron process or a two-electron process [109,110]. The mechanism(s) of oxygen reduction processes on carbon surfaces have been severally reported [79,89,109,111–118], but there is no consensus among researchers on any of the proposed mechanisms, the rate-determining step (RDS), or the adsorbed intermediate(s) involved in the electrochemical reduction of oxygen on carbon surfaces [115]. Nevertheless, there appears to be a consensus that the process involves the adsorption of oxygen and/or superoxide. The review article by Šljukić et al. [115] contains details

on the most prominent mechanisms proposed for oxygen reduction on carbon surfaces. Oxygen reduction in aqueous media is reported to be influenced by the test media and the nature of the electrode material [119,120]. Morcos and Yeager [121] reported a Tafel slope of −120 mV/decade at 25 °C in alkaline media, with a stoichiometric number of 2 for the overall two-electron reduction of $O_2$ to $HO_2^-$ on pyrolytic graphite surfaces. For the reduction of oxygen to $HO_2^-$ and $OH^-$ on glassy carbon, Zhang et al., [122] reported a Tafel slope of −60 mV/decade and a stoichiometric number of 1. Drawing on these works [121,122], Yeager [110] concluded that there are apparent mechanistic differences between $O_2$ reduction on pyrolytic graphite surfaces and glassy carbon. Major factors reportedly affecting the reactivity of carbon surfaces are mainly the microstructure of the carbon material [42,123–125], the cleanliness of the electrode surface [77,126,127], and the types of functional groups present on the surface [128–134]. This possibility of differences in the mechanism of cathodic processes on different carbon surfaces is the main driving force behind the present research effort.

In a recent publication [51], carbon-fiber-reinforced polymer (CFRP), which is a component in some multi-material assemblies, was characterized in near-neutral aqueous chloride media in an effort to evaluate and understand its ability to sustain cathodic activities (oxygen reduction reactions) under cathodic polarization consistent with galvanic coupling to metals in multi-material combinations. Oxygen-diffusion-limited cathodic current densities in the range of 30 to 40 μA cm$^{-2}$ have been reported [73]. Discounting self-corrosion and assuming equal surface areas, such cathodic current densities (40 μA cm$^{-2}$) had been estimated to be capable of supporting corrosion rates of about 0.599, 0.436, and 0.132 mm yr$^{-1}$, respectively, or mass loss rates of 11.708, 3.223, and 2.85 gm$^{-2}$ d$^{-1}$, respectively, on zinc, aluminium, and iron, respectively, galvanically coupled to CFRP [73]. Efforts aimed at mitigating such cathodic activities at potentials relevant to galvanic coupling to metals by the use of a variety of inhibitors have been reported [66,67], with marginal success obtained with the surfactant: sodium dodecyl sulphate (SDS), $Na_3PO_4$, lanthanum acetate, and inhibitor combinations comprising sodium nitrate with benzotriazole and cerium acetate with benzotriazole. The reported marginal suppression of cathodic activity on CFRP in the presence of these inhibitors was attributed to two mechanisms. Both mechanisms result in some reduction in the electrochemically active area, either via the interaction of inhibitor molecules with the CFRP surface, most probably by adsorption, and/or oxygen diffusion to the CFRP surface due to high-local-pH-triggered precipitation of inhibitor compounds on the CFRP surface [73].

Since the electrochemical activity of CFRP is attributable to its carbon content (and/or carbon fraction) in the composite in the form of carbon fibers, it can be argued that electrochemical data acquired from various carbon surfaces can be valuable for the design of high-performance multi-material combinations containing CFRP and optimized for both specific strength and the reduction in the electrochemical activity of the CFRP towards oxygen reduction reactions (ORR). To test this hypothesis, the electrochemical response of CFRP with 65% carbon fiber content was compared with electrochemical data acquired from glassy carbon and highly ordered pyrolytic graphite (HOPG), and the results are presented herein. Since the CFRP surface is a composite surface comprised of carbon fibers (of calculable surface coverage) and the polymer matrix and since the carbon fibers are deemed to be the conductive constituent (and plausibly the electrochemically active surface), its electrochemical response is compared with those of other carbon surfaces such as glassy carbon (GC) and highly ordered pyrolytic graphite (HOPG) with a known surface area. By comparison and analysis of the current profile and total quantity of charges passed (from chronoamperometric tests) across the CFRP–, GC–, and HOPG–solution interfaces under cathodic polarization during a fixed time, these carbon surfaces are ranked with respect to their plausible abilities to support cathodic processes and hence the concomitant anodic dissolution of coupled metal(s) in galvanic and/or multi-material combinations.

This work is focused on ORR, which is the predominant reaction at the cathodic potential ranges that CFRP is polarized to, on galvanic coupling to most metals in multi-material assemblies that are now common in the aerospace and transport industries. To mitigate the galvanic corrosion of coupled metallic components in such hybrid structures, it is desirable to engineer coupling carbon surfaces to have a lower response to ORR. The present work demonstrates and quantifies the implications and magnitude of the technological challenge arising from the well-known differences in the electrochemical activity of basal and edge carbon sites of carbon materials on corrosion mitigation in multi-material combinations. It provides quantitative data that clearly indicates that, in spite of its lower carbon content (about 65%), CFRP is more electrochemically active with regards to ORR compared to carbon surfaces with a greater surface area (glassy carbon and HOPG). The data generated in this study can serve as baselines for evaluating research efforts at modifying carbon-reinforced composite surfaces to achieve reduced response to ORR.

## 2. Materials and Methods

### 2.1. Materials Preparation

Pultruded CFRP rods of diameters 1 and 8 mm with 65% fiber content made of Tenax Total GmbH HT 24 K carbon fiber and epoxy vinyl matrix material were obtained from Modulor Material. These were cut to the required lengths and mounted in a polymeric resin (Epokwick) after attaching an insulated wire at one end for electrical contact. Before each test, a fresh surface was exposed by wet-grinding progressively with silicon carbide paper of grit sizes 240, 360, 400, 500, 600, and 1200, washing in between with copious amounts of distilled water. A glassy carbon electrode was prepared from glassy carbon samples with a cross-section of 10 × 10 mm that were mounted in polymeric resin (Epokwick) as for CFRP above, and the surfaces were prepared for testing using the same polishing procedure employed for CFRP. For tests with highly ordered pyrolytic graphite (HOPG), an HOPG sample of dimensions 10 × 10 × 2 mm was mounted on a conductive substrate and fresh surface-cleaved prior to each new test, with a test surface area of ≈ 0.28 cm² demarcated using an O-ring of internal diameter ≈ 5.5 mm and an in-house fabricated test rig.

### 2.2. Test Procedures

Electrochemical tests were carried out using an Autolab PGSTAT302N potentiostat, employing the 3-electrode method with a saturated calomel reference electrode, platinum wire as the counter electrode, and CFRP as the working electrode.

#### 2.2.1. Open-Circuit Potential Measurements

Open-circuit potentials of the CFRP, glassy carbon (GC), and highly ordered pyrolytic graphite (HOPG) in the 50 mM NaCl test solutions were monitored at 60 s intervals immediately after immersion until 10 h at least 3 times respectively.

#### 2.2.2. Potentiodynamic Tests

Triplicate potentiodynamic tests were carried out on CFRP, glassy carbon (GC), and highly ordered pyrolytic graphite (HOPG) respectively, in quiescent 50 mM NaCl test solutions with a potential step size of 2.44 mV and a scan rate of 1 mV s⁻¹ after 1 h immersion at OCP (without any conditioning), from 20 mV anodic of OCP towards cathodic potentials up to −2500 mV$_{SCE}$.

#### 2.2.3. Cyclic Voltammetry

Cyclic voltammetric tests were carried out on each of CFRP, glassy carbon (GC), and highly ordered pyrolytic graphite (HOPG) respectively, using a step size of 2.44 mV and a scan rate of 50 mV s⁻¹ from −1000 to +1000 mV$_{SCE}$ (from cathodic to more anodic potentials).

### 2.2.4. Chronoamperometric Measurements

Hextuple chronoamperometric tests were carried out on each of CFRP, glassy carbon (GC), and highly ordered pyrolytic graphite (HOPG) at a polarization of −1000 mV$_{SCE}$ in quiescent 50 mM NaCl for 1 h with current sampled every 0.5 s.

### 2.2.5. Electrochemical Impedance Spectroscopy

At OCP, electrochemical impedance spectroscopy was performed on CFRP, glassy carbon (GC), and highly ordered pyrolytic graphite (HOPG) using a single sine wave with a amplitude of 10 mV (RMS) in the frequency range of 100,000 Hz to 10 mHz. To check the effect of time on the impedance spectra, EIS tests were performed sequentially after 0.083 (5 min), 0.5, 1, 2, 3,4, 5, and 6 h of immersion.

### 2.2.6. Confocal Raman Spectroscopy

Raman spectra were acquired from the respective carbon surfaces before and after polarization using a WITec Alpha 300 RA+ confocal Raman system (WITec, Ulm, Germany) using a 532 nm laser and a laser power of 1 mW.

### 2.2.7. Scanning Electron Microscopy and Energy-Dispersive X-Ray Spectroscopy (EDXS)

The surfaces of the respective carbon surfaces before and after 1 h cathodic polarization in 50 mM NaCl at −1000 mV$_{SCE}$ were studied using a Hitachi TM4000Plus tabletop scanning electron microscope with an energy-dispersive X-ray spectroscopy (EDXS) capability provided by a Bruker Quantax 75 EDS system. The elemental mapping of similar surface areas of the respective samples' surfaces enabled the acquisition of comparative surface compositions of the carbon and oxygen on the respective carbon surfaces. Since the CFRP presents a composite surface, in addition to surface mapping, point elemental analysis on the carbon fibers on the CFRP surface was carried out to enable a more realistic comparison of the carbon surface composition before and after polarization.

## 3. Results

### 3.1. Comparison of the Open-Circuit-Potential Profiles of Carbon Surfaces in 50 mM NaCl

A comparison of the repeated measurements of the OCP profiles of the three different carbon surfaces (Figure 1) shows that the OCP profiles of CFRP and glassy carbon are very similar, but there is a significant difference in the OCP profiles of HOPG, which is predominantly anodic compared to the other carbon surfaces in the test media. This is principally attributed to poor electrode kinetics of the HOPG basal plane surface. However, for metallic surfaces, such an anodic shift in the evolution of the OCP profiles of HOPG would have been suggestive of a passivated surface and leads us to suspect the possibility of some passivation of the freshly cleaved HOPG surface. Some reports [102,103] appear to suggest that carbon surfaces can be passivated, particularly in acidic media, in a manner similar to that observed with some metallic materials, apparently due to the formation of a covering oxide layer (oxidized debris). Granted that, the nature of metallic surfaces and carbon surfaces differ, however, similar changes can occur on carbon surfaces on polarization. On polarization in aqueous media, the surface of carbon can be changed by (a) changes in the functional groups attached to carbon atoms [97,135], (b) electrochemically induced exfoliation of graphite/graphene layers (most probable in surfaces with a predominance of edge sites) [136,137], (c) erosion of graphite layers and creation of ledges, etc. (most probable in surfaces with a predominance of basal sites) [138–140], and (d) deposition of surface films on the carbon surface [141,142].

This position is further supported by the recent report of Patel et al. [143,144]. The earlier held traditional consensus was that (a) the basal surface of HOPG is characterized by very poor electrode kinetics [145] or even nil electroactivity [42,64,146–152] compared to edge plane graphite for a wide range of redox couples and (b) the intersection of step edges with the basal surface is essentially responsible for all of the sites for electron

transfer (ET) for a range of redox couples [33,42,64,147,153,154–156]. Contesting these points, Patel et al. [153,154] argued strongly and with some merit that freshly cleaved pristine HOPG is much more electrochemically active than previously thought (in the step-edge model of HOPG activity) but also that the HOPG basal surface readily passivates in a variety of ways.

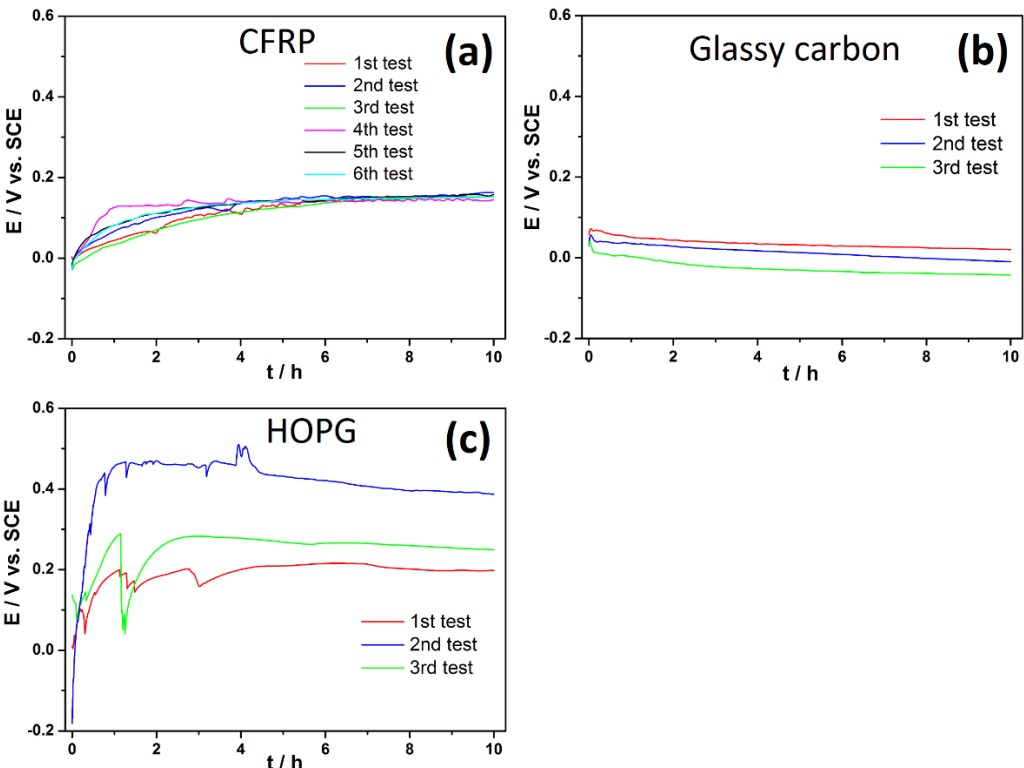

**Figure 1.** Open–circuit–potential profiles of (**a**) CFRP, (**b**) HOPG, and (**c**) glassy carbon in a 50 mM NaCl solution.

*3.2. Comparison of Cathodic Potentiodynamic Polarization Curves of Carbon Surfaces in 50 mM NaCl.*

From the potentiodynamic polarization curves for the three carbon surfaces (Figure 2), it can be observed that, whereas there is a marked resemblance in the cathodic polarization curves of CFRP and glassy carbon, that of HOPG is markedly different. The major difference in the cathodic polarization curves of HOPG in the test media compared to CFRP and GC (Figure 2d) is the absence of a clearly defined oxygen-diffusion-limited plateau region. For HOPG, oxygen reduction appeared to proceed without transport constraints on its surface until hydrogen evolution became the dominant cathodic reaction (at potentials ≤ −1500 mV$_{SCE}$ for all three carbon surfaces studied). The absence of a defined oxygen-diffusion-limited plateau region in the polarization curve for HOPG is indicative of very sluggish oxygen reduction kinetics on HOPG surfaces, which ensures that oxygen reduction does not come under mass transport (i.e., diffusion) control on HOPG surfaces. This phenomenon can be attributed to the preponderance of the less electrochemically active basal sites in the plane of the highly ordered pyrolytic graphite sample tested and apparently lends support to the "step-edge model" of HOPG activity.

In addition, for HOPG, the potentiodynamic curves (Figure 2d) indicate that the oxygen reduction reaction kinetics on its cleaved surface (basal plane) are slower compared to glassy carbon and CFRP. Morcos and Yeager [121] reported inhibited oxygen reduction and peroxide oxidation on the cleaved surfaces (basal planes) of the oriented graphite relative to edge surfaces in their studies on the kinetics of the oxygen–peroxide couple on pyrolytic graphite. Interestingly, unlike for glassy carbon and CFRP, the polarization

curve for HOPG shows two different slopes in the kinetics-controlled region of the curve (from about 0 mV$_{SCE}$ to about −1250 mV$_{SCE}$). Specifically, at potentials more negative (i.e., more cathodic) than −500 mV$_{SCE}$, the slope of the cathodic curve for HOPG (Figure 2d) is observed to change significantly before reaching the diffusion-controlled region around −1250 mV$_{SCE}$. This change in slope is indicative of a change in the predominant reduction mechanism or reaction for HOPG [157]. We attribute the first slope (moving from OCP and terminating around −500 mV$_{SCE}$) to be related to the predominant reduction of $O_2$ to $H_2O_2$, and the second slope to the predominant reduction of $H_2O_2$ to $H_2O$. These attributions find support from Qiang et al. [158], who in their studies of the electrochemical generation of hydrogen peroxide from dissolved oxygen in acidic solutions using graphite electrodes reported that the optimal potential for the electro-generation of $H_2O_2$ is a cathodic potential of −500 mV$_{SCE}$, that the decomposition of $H_2O_2$ is slow in the absence of metal ions, and that significant $H_2O_2$ self-decomposition occurs only at high pH (pH > 9) and at high temperatures (>23 °C). In addition, with consideration of the lower electrochemical activity of the HOPG basal plane, a recent report [51] indicates that the polarization of the HOPG-cleaved surface to −500 mV$_{SCE}$ is unlikely to raise the local pH beyond pH 9 so that $H_2O_2$ persists in solution for its subsequent electroreduction to water.

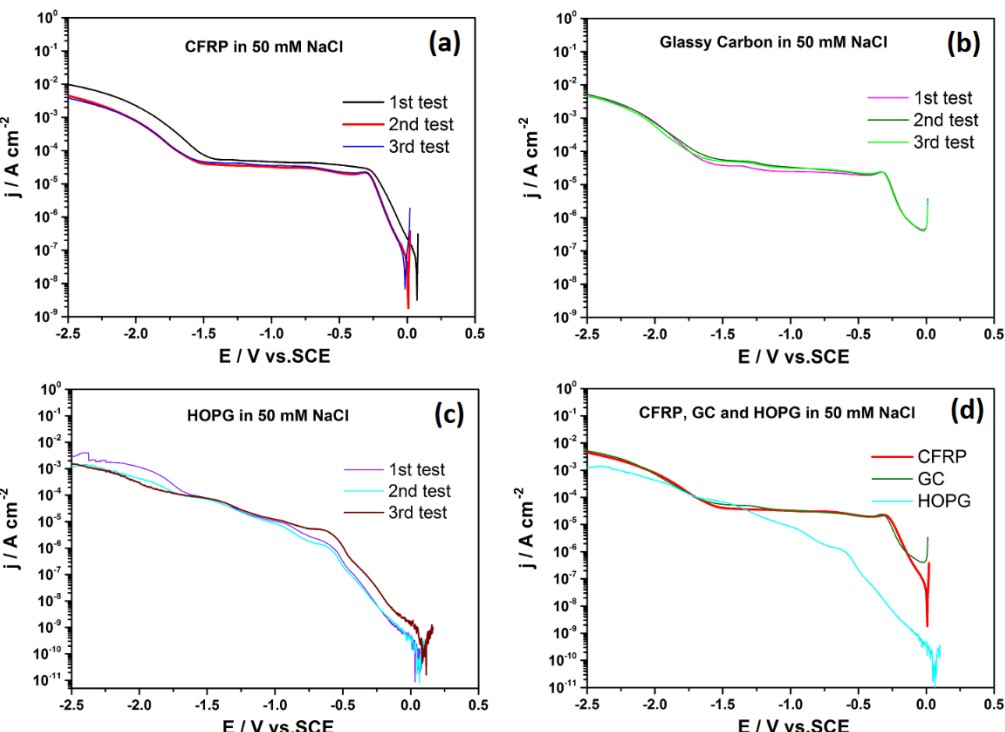

**Figure 2.** Comparison of cathodic potentiodynamic polarization curves of CFRP (**a**), and glassy carbon (**b**) HOPG (**c**), CFRP + GC+HOPG (**d**), in 50 mM NaCl acquired at scan rates of 1 mV s$^{-1}$.

### 3.3. Comparison of Cyclic Voltammograms in 50 mM NaCl.

The cyclic voltammograms of the three carbon surfaces (Figure 3) in solutions at different pH further confirms the low electrochemical activity of HOPG compared to other carbon surfaces and its insensitivity to pH in the pH range studied. For CFRP and glassy carbon, their cyclic voltammograms were marked by a prominent cathodic peak around −350 mV$_{SCE}$ at pH ≈ 7, 10, and 12. This peak around −350 mV$_{SCE}$ is attributed to a two-electron reduction of oxygen to $HO_2^-$ and $OH^-$ (Eo ≈ −309 mV$_{SCE}$) [159]. However, at pH ≈ 12, a second peak is observed in the cyclic voltammogram of glassy carbon around −700 mV$_{SCE}$, which we infer to be most probably linked to the reduction of $H_2O_2$ produced at less negative potentials (less cathodic potentials) [159]. Insights from the analysis of the cathodic polarization curves for HOPG (Figure 2d) indicate that $H_2O_2$ production is likely

to be more predominant in the potential range from 0 mV_SCE to −500 mV_SCE. The absence of this second peak on CFRP at the same pH (pH ≈ 12) is inferred to be indicative of some differences in the mechanism(s) of oxygen reduction on glassy carbon and CFRP surfaces. Earlier works [159–161] reported two peaks in the polarization curves of glassy carbon and polished pyrolytic carbon in oxygen-saturated alkaline solutions and attributed them to oxygen reduction on two different surface sites.

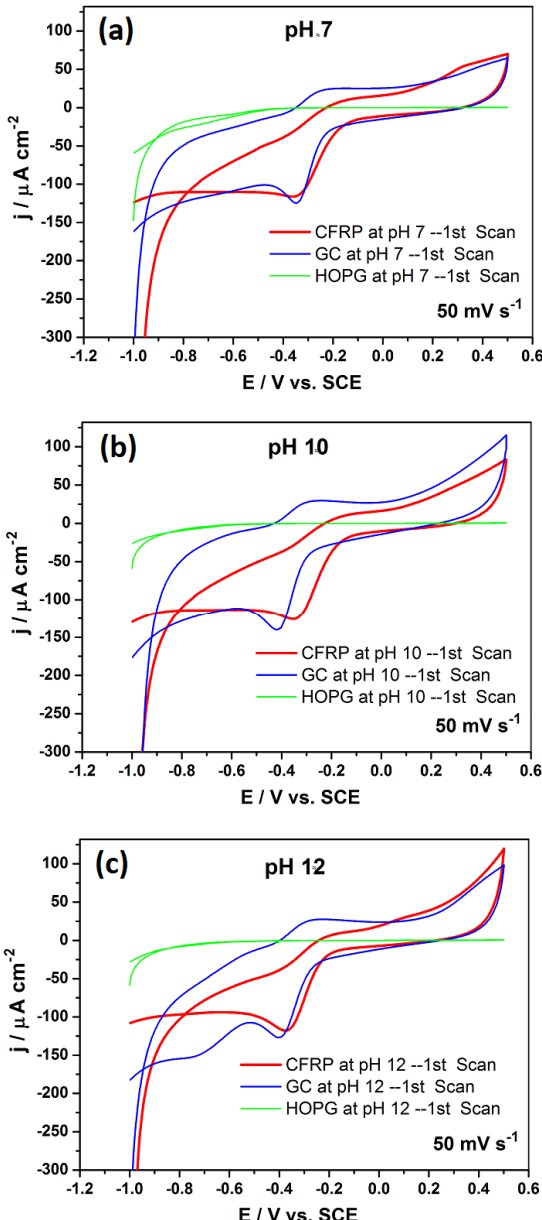

**Figure 3.** Comparison of cyclic voltammograms of CFRP, HOPG, and glassy carbon in 50 mM NaCl adjusted to pH 7 (**a**), 10 (**b**), and 12 (**c**) respectively, and at scan rates of 50 mV s⁻¹.

*3.4. Comparison of Electrochemical Impedance Spectra in 50 mM NaCl*

The impedance spectra of the three respective carbon surfaces present a single time constant, as can be seen from the Nyquist plots (Figure 4) and the Bode plots (Figure 5).

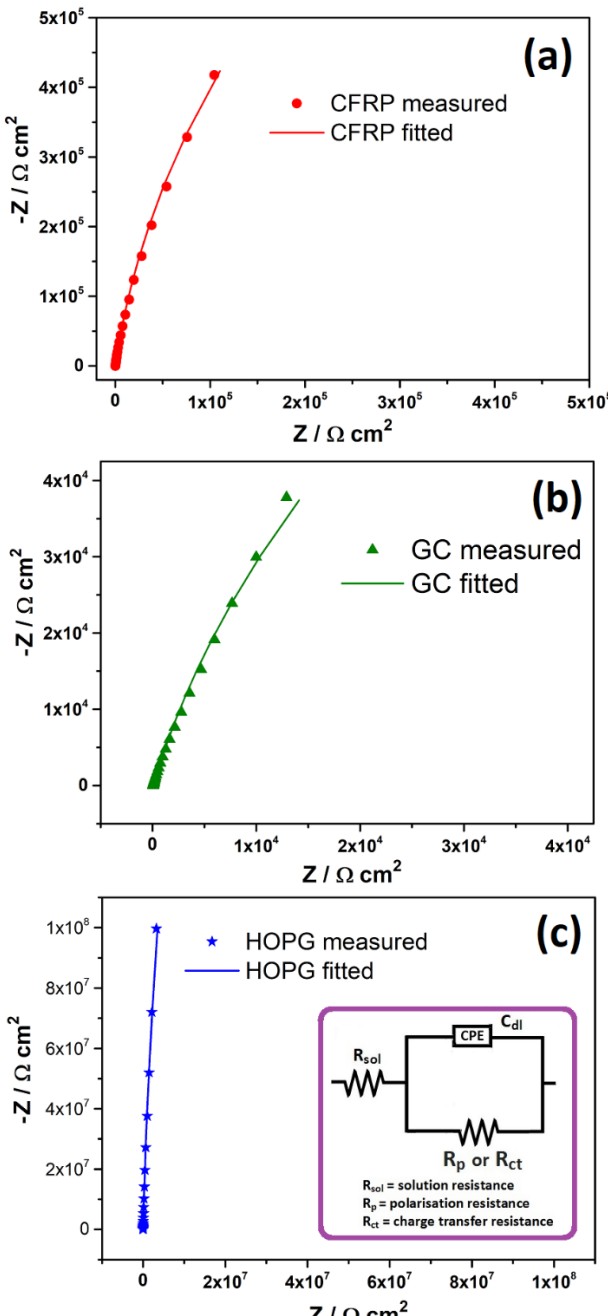

**Figure 4.** Nyquist plots of CFRP (**a**), glassy carbon (**b**), HOPG (**c**) in 50 mM NaCl adjusted to pH 7 after 1−h immersion at OCP (inset in c is the equivalent circuit used to fit the spectra).

The impedance spectra were fitted using the one-time constant equivalent circuit (inset in Figure 4c). In this equivalent circuit, a constant-phase element (CPE) is used instead of a capacitance to simulate the interface capacitance ($C_{dl}$) generated on the carbon surfaces in order to account for surface inhomogeneities that can lead to a distribution of relaxation time constants and hence deviations from ideal capacitor behavior [162–164]. The CPE is expressed and related to the capacitance according to Equation (1);

$$\frac{1}{Z} = Y = Y_o \, (j\,\omega)^n \tag{1}$$

where Z is the impedance of the CPE, and $Y_o$ and n are the characteristic parameters of the CPE, j is a mathematical operator (imaginary number equal to square root (−1)) associated with the phase information (imaginary component of the circuit), $\omega$ is the angular frequency, and the exponent n is valued ($0 < n < 1$); exponent $n = 1$ for the ideal capacitor response.

Capacitances were calculated from impedance data using the relation (Equation (2)) proposed by Brug et al. [165].

$$C = Y_o^{\frac{1}{n}} \left[\frac{1}{R_s}\right]^{\frac{n-1}{n}} \tag{2}$$

where $Y_o$ is the fitted value of CPE element, $R_s$ is the resistance in series with a CPE element (in this case, the solution resistance), and n the CPE exponent.

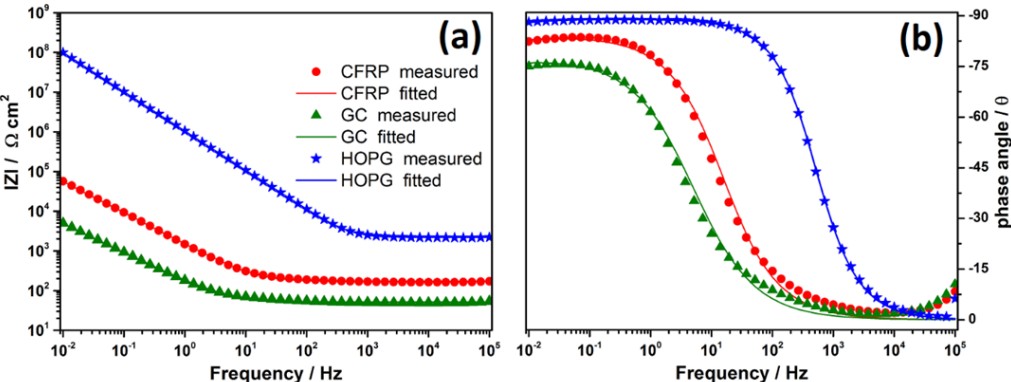

**Figure 5.** Bode plots of CFRP, HOPG, and glassy carbon in 50 mM NaCl adjusted to pH 7 after 1−h immersion at OCP presented as (**a**) IZI and (**b**) phase angle plots (inset in Figure 4c is the equivalent circuit used to fit the spectra).

From the time evolution of low-frequency impedance (at $10^{-2}$ Hz) (Figure 6a), it can be observed that HOPG presents the highest global impedance of all three carbon surfaces studied, even after accounting for the differences in the "solution resistance". A high global impedance compared to other carbon surfaces is indicative of a lower electrochemical response of HOPG to oxygen reduction, which is sustained during the 6 h test duration By fitting the acquired impedance spectra using the equivalent circuit (Figure 4c inset), it was possible to de-convolute the global impedances into their respective resistive components (charge transfer resistance, Rct) and their capacitive component (the double layer capacitances) and to follow their evolution as a function of time for each carbon surface (Figure 6b,c). From Figure 6b, it is obvious that the polarization resistance is markedly and consistently higher for HOPG than for CFRP or glassy carbon, ranking in the order; HOPG >> CFRP > GC, and indicative of a higher resistance to charge transport across the HOPG–solution interface. Contrary to this hypothesis and in agreement with capacitance values reported by earlier workers for HOPG (1.7 to 2.8 μF cm$^{-2}$ for ZYA-HOPG [143], 0.7 μF cm$^{-2}$ for mechanically cleaved HOPG [46], 1.9 μF cm$^{-2}$ for low-defect HOPG [166], and 3.4 to 7.1 μF cm$^{-2}$ for the ZYH grade of HOPG, and the high-value range (>3 μF cm$^{-2}$) deemed to be indicative of higher defect concentrations [33,156,166,167]), markedly low double-layer capacitances of 1.39 to 1.77 μF cm$^{-2}$ were obtained in the present work for HOPG compared to 19.5 to 34.4 μF cm$^{-2}$ for CFRP and 13.6 to 85.5 μF cm$^{-2}$ for glassy carbon. Considering the HOPG surface to be predominated by basal-plane graphite sites, the capacitance values obtained for HOPG compared to GC and CFRP is consistent with McCreery and co-workers' report [166] that edge-plane graphite exhibits

capacitance values that can be more than 30-fold higher than that of basal-plane graphite. It is worth noting that recent studies postulate that capacitance is a weak indicator of the surface quality of carbon surfaces [168,169].

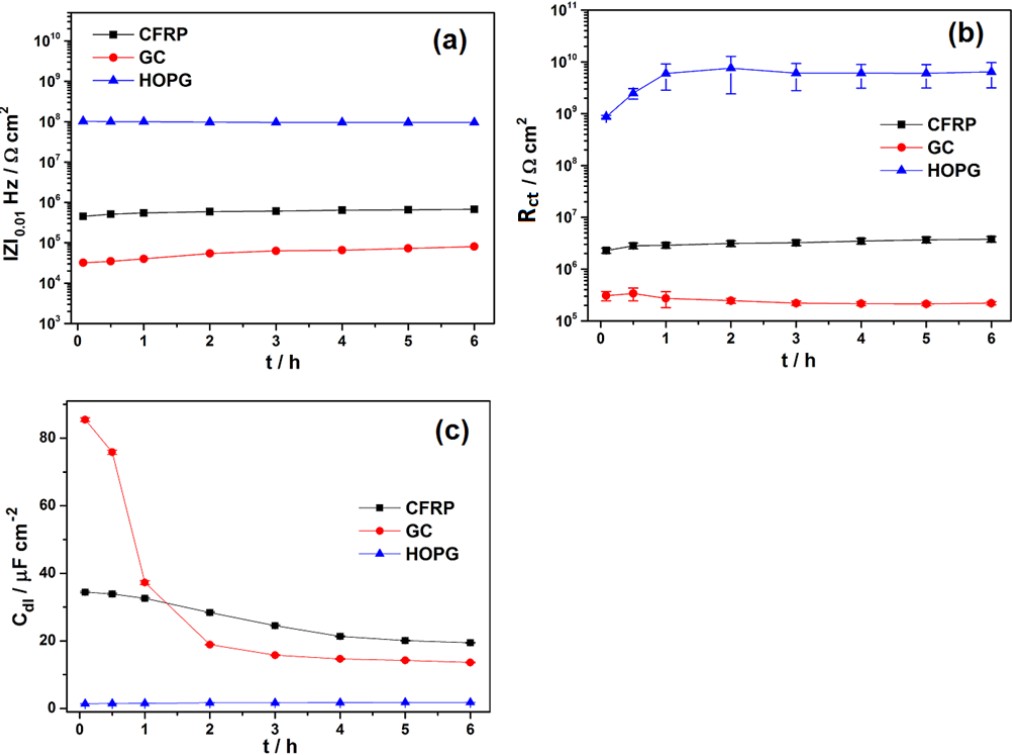

**Figure 6.** Comparison of the time evolution of measured and fitted parameters from EIS data from CFRP, HOPG, and glassy carbon in 50 mM NaCl adjusted to pH 7, (**a**) low-frequency impedance at 0.01 Hz (IZI$_{0.01 Hz}$), (**b**) charge transfer resistance (Rct), and (**c**) interface capacitance (C$_{dl}$) (error bars indicate fitting errors).

In general, the capacitance values decreased slowly with immersion time in all three carbon surfaces, with the exception of glassy carbon, for which the decrease was fast during the first two hours. Randin and Yeager [170] reported a similar decrease in capacitance with time. The observation (from Figure 6b) that, during the first 1 h, in which the decrease in capacitance was most prominent for glassy carbon and CFRP surfaces (regarded to have more edge sites), coincided with a progressively increasing trend in the parallel resistance (Rct in the inset of Figure 4c) leads us to conclude that this is most likely due to some surface modification(s) on the carbon surfaces that result in an increase in surface area, which might be most prominent on the glassy carbon surface. The increased surface area is bound to increase the value of A (area of plates) in Equation (3), which is directly proportional to the capacitance and is bound to result in a reduction in capacitance as observed. This scenario could explain both the observed decrease in capacitance with time until the surface stabilizes, as well as the capacitance and charge transfer resistance(s) stabilizing to lower and more stable values at extended immersion times (t → ∞).

$$C = \frac{\varepsilon_o\ \varepsilon_r\ A}{d} \qquad (3)$$

where C is capacitance, $\varepsilon_o$ is the permittivity of free space (8.854 × 10$^{-12}$ F m$^{-1}$), $\varepsilon_r$ is the dielectric constant (of the electrolyte), and A is the distance between the two plates (electrode surfaces).

### 3.5. Comparison of Chronoamperomteric Profiles in 50 mM NaCl

Hextuple chronoamperometric measurements were carried out on CFRP, glassy carbon (GC), and highly ordered pyrolytic graphite (HOPG) in quiescent 50 mM NaCl by polarizing the samples to −1000 mV$_{SCE}$ for 1 h while measuring the current in 0.5 s intervals.

Averaged values of the six chronoamperometric profile measurements are presented in Figure 7, which shows that CFRP was most electrochemically active towards ORR under test conditions. The individual repeated chronoamperometric profiles for each carbon surface are presented in Supplementary Material Figure S1. By the integration of the currents measured over the test duration (1 h), the quantity of charge passed on each test for each carbon surface was determined, normalized using the exposed sample surface to obtain the specific charge passed, and presented as Supplementary Materials Figure S2 and Table S1. Using GraphPad version 6, this data was statistically analyzed using one-way analysis of variance (ANOVA) and Bonferroni's multiple comparison test with significance set at $p < 0.05$. The results from the statistical analysis (inset in Figure 7 and Supplementary Material Table S2) indicate that the averages (CFRP, 129.52 mC cm$^{-2}$; glassy carbon, 89.95 mC cm$^{-2}$; HOPG, 60.77 mC cm$^{-2}$) of the charges passed through each carbon surface are statistically significant. Since during galvanic corrosion with coupled metals (and hence, cathodic polarization), charges need to be passed between the cathode (example CFRP) and the anode (metallic material) to bring about enhanced anodic metal dissolution, a measure of the quantity of charges passed across each carbon surface under same impressed cathodic polarization can be taken as an indicator of the magnitudes of cathodic activities supportable by each carbon surface. Employing this as a ranking criterion, the propensities of the three carbon surfaces (CFRP, GC, and HOPG) to support cathodic activities that can lead to anodic metal dissolution on galvanic coupling to metallic materials can be visualized (as inset in Figure 7).

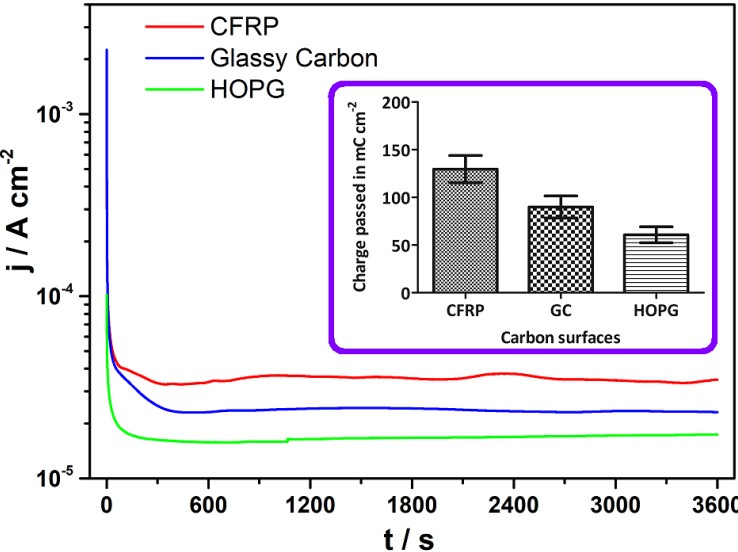

**Figure 7.** Averaged chronoamperometric profile from hextuple measurements of each carbon surface (Inset is the output from the statistical comparison of charges passed during 1−h polarization of CFRP, HOPG, and glassy carbon in 50 mM NaCl at −1000 mV$_{SCE}$ from hextuple chronoamperometric tests, respectively). (Error bars indicate standard error of means (SEM)).

### 3.6. Confocal Raman Studies of Carbon Surfaces

From Figure 8, it can be observed that, generally, the polarization of the respective carbon surfaces had little effect on the Raman spectra of the carbon surfaces, as no new peaks were observed after polarization compared to the corresponding unpolarized samples. However, it can be observed that, for glassy carbon (Figure 8b), all the peaks

observed in the Raman spectra of the unpolarized sample became much more prominent (manifesting higher Raman intensities) after polarization. In contrast, for HOPG, a marginal decrease in the Raman intensity of all the peaks present in the Raman spectra of an unpolarized HOPG surface (freshly cleaved surface) was observed after polarization.

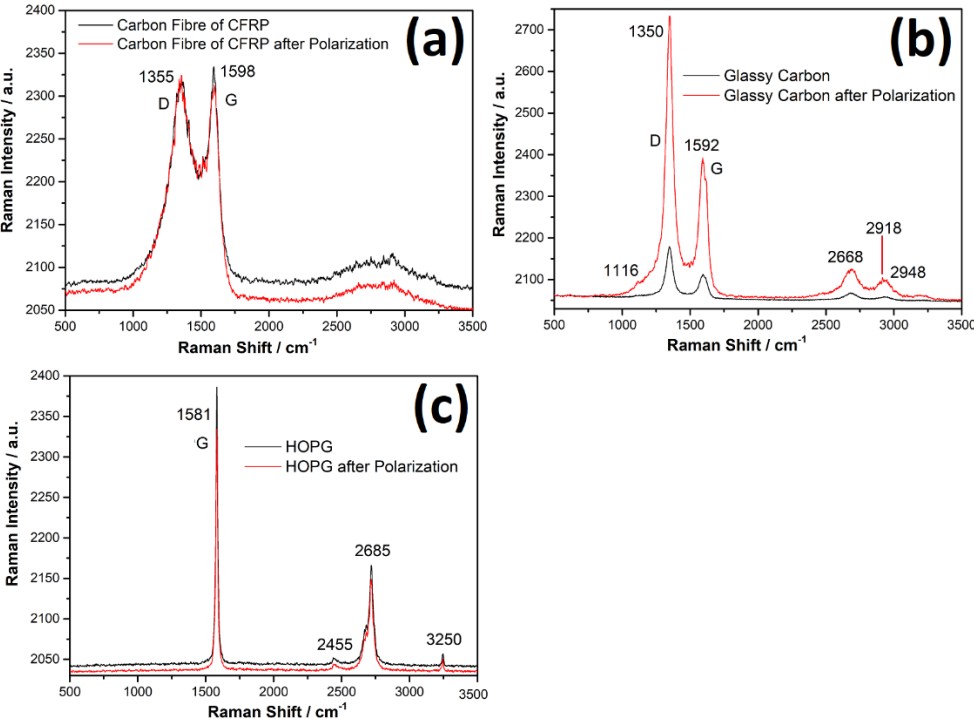

**Figure 8.** Confocal Raman spectra of (**a**) carbon fiber surfaces in CFRP, (**b**) glassy carbon surfaces, and (**c**) HOPG surfaces before and after 1 h polarization in 50 mM NaCl at −1000 mV$_{SCE}$, respectively.

### 3.7. Energy-Dispersive Spectroscopy of Carbon Surfaces

From the EDS data of the carbon surfaces in Table 1, it can be observed that the surface composition of each of the three carbon surfaces changed after polarization compared with values measured in unpolarized samples. The oxygen content of the carbon surfaces (atom. %) increased after polarization. A very small increase in surface oxygen concentration was observed on the HOPG surface after polarization. Surface oxygen concentration of the carbon surfaces increased in the following order; CFRP > GC > HOPG. The observed trend of increased oxygen concentration on the post-polarized carbon surfaces is consistent with the ranking of their responses to ORR. Consequently, it is concluded that their respective responses to ORR can be significantly attributed to differences in their propensities to undergo surface modification under the test conditions. The differences in their respective abilities to undergo surface modification can in turn be linked to initial differences in the structural and chemical compositions of the respective carbon surfaces.

**Table 1.** Table of elemental compositions of carbon surfaces determined by energy-dispersive x-ray spectroscopy (EDS) before and after polarization in a 50 mM NaCl solution.

| Carbon Surface | Status | Elements | Atomic No. | Counts | Mass [%] | Mass Norm. [%] | Atom [%] | abs. Error [%] (1σ) | Rel. Error [%] (1σ) |
|---|---|---|---|---|---|---|---|---|---|
| CFRP | polished | C * | 6 | 50,588 | 95.25 | 95.25 | 96.39 | 4.12 | 4.33 |
| | polished | O * | 8 | 822 | 4.75 | 4.75 | 3.61 | 0.31 | 6.62 |
| | polished | C ** | 6 | 41,255 | 95.73 | 95.73 | 96.76 | 4.14 | 4.33 |
| | polished | O ** | 8 | 593 | 4.27 | 4.27 | 3.24 | 0.31 | 7.15 |
| CFRP | **polarized** | **C *** | **6** | **98,313** | **93.28** | **93.28** | **94.89** | **4.05** | **4.35** |
| | **polarized** | **O *** | **8** | **2392** | **6.64** | **6.64** | **5.07** | **0.38** | **5.72** |
| | **polarized** | **C **** | **6** | **83,433** | **93.04** | **93.04** | **94.68** | **4.04** | **4.35** |
| | **polarized** | **O **** | **8** | **2135** | **6.96** | **6.96** | **5.32** | **0.40** | **5.77** |
| Glassy Carbon | polished | C # | 6 | 61,582 | 97.61 | 97.61 | 98.20 | 4.19 | 4.29 |
| | polished | O # | 8 | 463 | 2.37 | 2.37 | 1.79 | 0.18 | 7.61 |
| | polished | C ## | 6 | 76,278 | 97.98 | 97.98 | 98.47 | 4.20 | 4.28 |
| | polished | O ## | 8 | 482 | 2.02 | 2.02 | 1.53 | 0.15 | 7.64 |
| Glassy Carbon | **polarized** | **C** | **6** | **55,184** | **96.66** | **96.66** | **97.54** | **4.18** | **4.33** |
| | | **O** | **8** | **566** | **3.09** | **3.09** | **2.34** | **0.22** | **7.23** |
| | | **Na** | **11** | **266** | **0.19** | **0.19** | **0.10** | **0.03** | **15.56** |
| | | **Cl** | **17** | **201** | **0.07** | **0.07** | **0.02** | **0.01** | **17.09** |
| HOPG | Freshly cleaved | C | 6 | 266,167 | 99.65 | 99.65 | 99.74 | 4.23 | 4.25 |
| | | O | 8 | 276 | 0.35 | 0.35 | 0.26 | 0.04 | 11.97 |
| HOPG | **polarized** | **C** | **6** | **125,020** | **99.78** | **99.78** | **99.84** | **4.25** | **4.25** |
| | | **O** | **8** | **76** | **0.21** | **0.21** | **0.16** | **0.06** | **26.78** |

[* denotes elemental data values from EDS mapping of CFRP surface comprised of carbon fibers and the epoxy matrix. ** denotes elemental data values from point analysis on carbon fiber on the CFRP surface. # and ## denote the values from the first and second measurements of glassy carbon, respectively, after polishing. Data from polarized samples are in bold].

## 4. Discussion

In the chronoamperometric measurements in quiescent 50 mM NaCl solutions in Figure 7 (and supplementary materials Figures S1 and S2), the charge passed per square centimeter of the exposed surface during 1 h polarization of CFRP, glassy carbon, and HOPG was in the range of ≈115–147, 80–105, and 51.4–73.6 mC cm$^{-2}$, respectively, while the averaged values were ≈129.5, 90, and 60.8 mC cm$^{-2}$, respectively. The fact that there was no overlap in the values measured for the different carbon surfaces in six different measurements on each surface confirms clear differences in the reactivity of the different carbon surfaces under impressed cathodic polarization and under quiescent conditions.

Interestingly, it is observed that the charges passed during 1 h for CFRP are higher than those for glassy carbon in the chronoamperometric tests (Figure 7 and Supplementary Material Figure S1), while their cathodic current densities are similar in the polarization curves (Figure 2). These can be clarified by repeated chronoamperometric testing at different agitation rates in order to control oxygen transport conditions. However, such dynamic tests (under forced oxygen mass transport conditions) are inconsistent with our current focus on the galvanic-induced corrosion of metallic members of multi-material structures containing carbon-based material components under quiescent electrolyte conditions and hence, are beyond the scope of the present study.

The comparison of the quantity of charges passed across CFRP, GC, and HOPG during 1 h under quiescent conditions indicates that CFRP, in spite of its lower fraction of carbon surface, was the most electrochemically active at the impressed cathodic polarization, followed by glassy carbon (GC) and then HOPG. This order of the electrochemical

activity of the "carbon surfaces" (CFRP > GC > HOPG) is attributed to differences in the distribution of the more electrochemically active edge plane sites on the different carbon surfaces. This ranking, in which HOPG manifests the least electrochemical activity, is consistent with the report of Cline et al. [171], who reported anomalously slow electron transfer kinetics on HOPG surfaces.

The very low (essentially inert) response of the HOPG surface to ORR can be attributed to the low content of the edge plane sites, which on HOPG surfaces emanate from surface defect sites. According to previous works [64,172,173], a given HOPG surface consists, on average, of 1–10% edge-plane sites. while the remaining surface (90–99%) consists of much less electrochemically active basal-plane sites. The equally low capacitance values observed are consistent with earlier reports (Table 2).

**Table 2.** Reported capacitance values for different carbon surfaces and sites.

| Material | Edge Plane Capacitance ($\mu F\ cm^{-2}$) | Basal Plane Capacitance ($\mu F\ cm^{-2}$) | Material Capacitance ($\mu F\ cm^{-2}$) | References |
|---|---|---|---|---|
| Graphene | $1.0 \times 10^5$ (nano-effects could be present) | 4 | | [65] |
| Graphite | 2.8 | 1.8 | | [174] |
| HOPG | | 0.81–1.0 | | [145] |
| Graphite | 1–3 | 50–100 | | [170,175,176] |
| Glassy carbon | | | ≥13 | [170] |
| Carbon felt | 47.8 | 3.2 | | [177] |
| Glassy carbon | | | 10–25 | [177] |
| Glassy carbon | | | 35 | [69] |
| Hydrogenated glassy carbon | | | 20 | [69] |
| HOPG | | | 6 | [69] |
| Diamond | | | 5 | [69] |
| CFRP | | | **19.5**–34.5 | This work |
| Glassy carbon | | | **13.6**–85.5 | This work |
| HOPG | | | **1.4**–1.8 | This work |

[In bold are more stable capacitance values obtained after 6 hours' immersion in this work].

According to McDermott et al. [145], glassy carbon has faster kinetics than the HOPG basal plane due to its higher edge plane density. The results from this work indicate that CFRP is more electrochemically active towards ORR than glassy carbon and much more electrochemically active than HOPG (basal plane sites). In addition, the rankings of electrochemical responses and parameters for glassy carbon and HOPG obtained in this work are also consistent with the ranking and (capacitance) values reported by Xu et al. [69].

Typical Raman spectra of carbon materials are often composed in most instances of at least three bands denoted as D (observed around 1350 $cm^{-1}$), G (observed around 1385 $cm^{-1}$), and 2D, often also denoted as G' (observed around 2700 $cm^{-1}$), respectively [178,179]. The D-band is attributed to the presence of defects (vacancies and dislocations in the graphene layer and at its edges) [178–181]. The G band has been attributed to the in-plane vibration of $sp^2$-hybridized carbon atoms [182–185] and the 2D or G' band to the number of graphene layers [186,187] in the carbonaceous material. Besides these, additional bands have been reported (D' around 1620 $cm^{-1}$, peak denoted as D* or G* around 2440 $cm^{-1}$, and other peaks denoted as D + D' or D + G and 2D' or G + D) by the deconvolution of the Raman spectra, and additional information on the carbonaceous material has been extracted by the comparison of the ratios of the intensities of these peaks

[179,181,188,189]. Employing a laser source with a wavelength similar to that used in this work (532 nm), Kaniyoor and Ramaprabhu [179] reported that the Raman spectra of graphite oxide (GO) present intense D and G bands and a flat 2D region and show that the D band is more intense than the G band (with $I_D/I_G$ = 1.1). In this work (Figure 8b), the $I_D/I_G$ ratios of glassy carbon increased from 1.03 in unpolarized samples to 1.14 in the polarized sample, which suggests the presence of graphite oxide after cathodic polarization in the test media. Both graphene oxide (GO) and reduced graphene oxide (rGO) are reported to be catalytic towards ORR by causing it to proceed via a two-electron mechanism at low overpotentials [190]. However, reduced graphene oxide (rGO) is much more catalytic toward ORR than graphene oxide (GO) [190,191]. According to Kauppila et al. [192], graphene oxide can be electrochemically reduced in an aqueous solution by applying cathodic polarization in the range of −711 mV vs. Ag/AgCl (≡−756 mV vs. SCE) at pH 2 to −1416 mV vs. Ag/AgCl (≡−1661 mV vs. SCE) at pH 12. The reduction of graphene oxide in this cathodic potential range has been corroborated by other workers [190,193–195]. Based on the fact that (a) the cathodic potential used in this work falls within the range reported for the graphene oxide (GO) electrochemical reduction to reduced graphene oxide (rGO) in an aqueous solution, (b) the presence of graphite/graphene oxide post anodization on glassy carbon can be inferred from the Raman spectra (Figure 8b), and (c) increased oxygen contents in all three carbon surfaces were detected by EDS analysis after cathodic polarization (Table 1), we postulate the likely presence of graphene oxide and/or reduced graphene oxide in situ, in varying proportions on the carbon surfaces during cathodic polarization. This can partly account for the observed differences in their respective responses to ORR under the test conditions.

The high electrochemical activity of CFRP demonstrated in this work can be linked to the production process of the carbon fibers and their uni-directional arrangement (in the samples used in this work), which favor the presence of more edge sites on the exposed (transverse) ends of the carbon fibers in the CFRP samples [64]. In light of this, it is obvious that the electrochemical activity of CFRP samples with multi-directional carbon fibers is likely to differ from the values reported herein. The intermediate electrochemical activity of glassy carbon is attributed to the fact that the glassy carbon surface is composed of a less steep distribution of the more electrochemically active edge sites and the less electrochemically active basal sites. The lower measured electrochemical activity of the HOPG is attributed to the predominance of the less electrochemically active basal plane sites [196] on the surface of the highly ordered carbon surface tested. It is noteworthy that, in spite of the differences in the electrochemical activity of the different carbon surfaces, the cathodic current densities (attributable to oxygen reduction processes) varied insignificantly for CFRP and glassy carbon in the polarization plots (Figure 2d). Based on of the values and trends of electrochemical parameters (primarily capacitance) obtained in this work for CFRP (19.5 to 34.5 $\mu F\ cm^{-2}$), glassy carbon (13.6 to 85.5 $\mu F\ cm^{-2}$), and HOPG (1.4 to 1.8 $\mu F\ cm^{-2}$), as well as values reported by Yuan et al., [65] for edge plane sites ($1.0 \times 10^5$ $\mu F\ cm^{-2}$) and basal plane sites (4 $\mu F\ cm^{-2}$), and by other workers (presented in Table 1) on carbon materials, we posit that the observed differences in electrochemical responses of these three carbon-rich surfaces to ORR is significant due to differences in the proportions of edge sites present on each carbon surface, as illustrated in Figure 9.

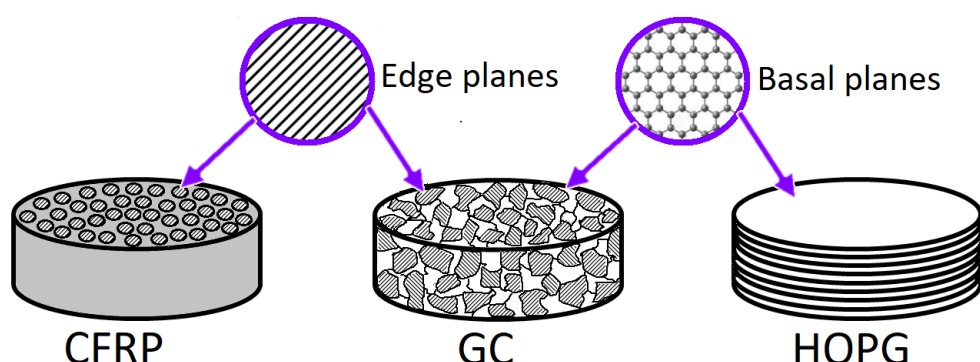

**Figure 9.** Schematic representation of the hypothesized differences in electrochemically active surfaces of the carbon surfaces studied.

## 5. Conclusions

A comparison of the electrochemical data of the studied carbon surfaces at potentials relevant to galvanic coupling to metals indicated that, at this cathodic potential, the CFRP surface is the most electrochemically active of the studied carbon surfaces, with the ranking of the surfaces following the order; CFRP > GC > HOPG. This ranking is attributed to differences in the ratios of the more electrochemically active edge sites compared to the less active basal sites on each carbon surface. Based on these results, it is concluded that the use of data acquired from non-composite ("homogenous") carbon surfaces to estimate the electrochemical activity of carbon-fiber-reinforced polymers (CFRP) in the design of corrosion-resistant multi-material assemblies is likely to lead to an under-estimation of the severity of the cathodic activities on the CFRP surface that support anodic dissolution of metals galvanically coupled to CFRP in such hybrid assemblies. When collecting such data from CFRP surfaces is difficult, data collected from the glassy carbon surface are advocated as a viable alternative based on the strength of trends observed in the potentiodynamic polarization and electrochemical impedance spectroscopy tests in this work. Engineering carbon-based surfaces to appear as dispersed micrometer-sized arrays of predominantly carbon edge plane sites can be a feasible technological approach for applications requiring fast oxidation rates for ORR. In contrast, for corrosion mitigation, in which the suppression of ORR is a goal, surface engineering to achieve a predominance of carbon basal sites can be a suitable solution. It is envisaged that these results will motivate future efforts at engineering CFRP surfaces that have a reduced response to ORR similar to that observed on HOPG, as this would result in more corrosion-resistant hybrid structures. In addition, the data from this work can be valuable as a baseline for ranking the performance of carbon surfaces engineered for reduced response to ORR.

**Supplementary Materials:** The following supporting information can be downloaded at: https://www.mdpi.com/article/10.3390/c9010007/s1, Figure S1: Hextuple chronoamperometic profiles of the 3 carbon surfaces during 1 h polarization at −1000 mV$_{SCE}$ in 50 mM NaCl; Figure S2: Comparison of charges passed during 1 h polarization of CFRP, HOPG, and glassy carbon in 50 mM NaCl at −1000 mV$_{SCE}$ in hextuple chronoamperometric tests; Table S1: Table of integrated charges passed in hextuple chronoamperomteric measurements; Table S2: Table of outputs from ANOVA tests for significance.

**Authors Contributions:** Conceptualization, S.U.O. and M.Z.; methodology, S.U.O.; validation, S.U.O., M.G.S.F., H.I.S.N., and M.Z.; formal analysis, S.U.O. and H.I.S.N.; investigation, S.U.O. and H.I.S.N.; resources, M.Z. and H.I.S.N.; data curation, S.U.O.; writing—original draft preparation, S.U.O.; writing—review and editing, M.G.S.F. and M.Z.; visualization, S.U.O.; supervision, M.G.S.F. and M.Z.; project administration, M.Z.; funding acquisition, M.Z. All authors have read and agreed to the published version of the manuscript.

**Funding:** S.U. Ofoegbu acknowledges Fundação para a Ciência e a Tecnologia (FCT) Portugal for the doctoral grant (SFRH/BD/75167/2010). Funding from FCT project: "Corrosion and Corrosion Protection in Multi-material Systems", (PTDC/CTM/108446/2008), and European FP7 project:

"Active PROtection of multi-material assemblies for AIRcrafts" ["PROAIR" (PIAPP-GA-2013-612415)] are acknowledged. This work was developed within the scope of the project CICECO-Aveiro Institute of Materials, POCI-01-0145-FEDER-007679 (FCT Ref. UID/CTM/50011/2013), financed by national funds through the FCT/MEC and, when appropriate, co-financed by FEDER under the PT2020 Partnership Agreement.

**Institutional Review Board Statement:** Not applicable.

**Informed Consent Statement:** Not applicable.

**Data Availability Statement:** The raw data generated in the course of this study and supporting reported results are publicly archived in the Mendeley data repository and can be accessed using the link: https://doi.org/10.17632/x8c3kpc49g.2.

**Acknowledgments:** Rui Sampaio and Alexandre C. Bastos both of the Department of Materials and Ceramic Engineering, CICECO-Aveiro Institute of Materials, Aveiro (Aveiro, Portugal), are gratefully acknowledged for graciously providing the glassy carbon samples used, Kiryl Yasakau of the Department of Materials and Ceramic Engineering, CICECO-Aveiro Institute of Materials, Aveiro (Aveiro, Portugal), is gratefully acknowledged for graciously loaning us the HOPG sample.

**Conflicts of Interest:** The authors declare no conflict of interest. The funders had no role in the design of the study; in the collection, analyses, or interpretation of data; in the writing of the manuscript; or in the decision to publish the results.

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
