# Peer review of "Comparison of the Electrochemical Response of Carbon-Fiber-Reinforced Plastic (CFRP), Glassy Carbon, and Highly Ordered Pyrolytic Graphite (HOPG) in Near-Neutral Aqueous Chloride Media"

_carbon, 2022_

Round 1

Reviewer 1 Report

Authors have studied and compared the electrochemical responses of CFRP, GCE and HOPG in aqueous chloride media. Though, all the experiments are well done and results are analysed properly. As author pointed out, most of the things are already found and reported in the literature. I really do not understand the purpose of this research and what is novelty in this study. Author should describe the main new findings clearly in the introduction section.

Major comments:

  1. how did you choose 50 mM NaCl without studying the other concentrations range to all your experiments. Ionic strength of the solution is also may affect the electro-oxidation or reduction of carbon surfaces.
  2. Fig. 8 is meaning less, if you don’t show the FE-SEM images of the surface after oxidation.
  3. It is required to show some spectrosocpic data (FT-IR) to confirm that surface has undergone oxidation or reduction.
  4. In this work, only electrochemical data are provided without any other supporting data from EDX, SEM, IR, contact-angle, etc.
  5. Introduction section should be updated and the following articles need to be cited, https://doi.org/10.1149/2754-2726/ac7c78https://doi.org/10.3389/fmats.2022.906838
  6. Minor English editing is required.

Author Response

Reviewer 1:

Reviewer 1: General Comment 1:

Authors have studied and compared the electrochemical responses of CFRP, GCE and HOPG in aqueous chloride media. Though, all the experiments are well done and results are analysed properly. As author pointed out, most of the things are already found and reported in the literature. I really do not understand the purpose of this research and what is novelty in this study. Author should describe the main new findings clearly in the introduction section.

Response to Reviewer 1, General Comment 1:

A paragraph has now been added to the introduction presenting the novelty of the present work.

Previous works in our laboratory in this system indicate that at cathodic polarization of -1000 mV vs. SCE ORR is the predominant cathodic reaction. This work is thus focused on ORR because this is the predominant reaction at the cathodic potential ranges CFRP is polarized to on galvanic coupling to most metals in multi-material assemblies now common in the aerospace and transport industries.  To mitigate galvanic corrosion of coupled metallic components in such hybrid structures, it is desirable to engineer coupling carbon surfaces to have lowered response to ORR. The present work demonstrates that among the 3 carbon surfaces investigated the CFRP surface manifests the greatest response to ORR. It is hoped that these results will motivate future efforts at engineering CFRP surfaces that have a reduced response to ORR similar to that observed on HOPG, as this would result in better corrosion-resistant hybrid structures.

The novelty of this work is that it demonstrates and quantifies the implications and magnitude of the technological challenge arising from the well-known differences in the electrochemical activity of basal and edge carbon sites in carbon materials in corrosion mitigation in multi-material combinations. It provides quantitative data which indicates clearly that in spite of its lower carbon content (about 65%) CFRP is more electrochemically active with regards to ORR compared to carbon surface of greater surface area (glassy carbon and HOPG). The data generated in this study can form baselines for evaluating research efforts at modifying carbon-reinforced composite surfaces to achieve reduced response to ORR.

 Reviewer 1, Major comments:

Reviewer 1, Major Comment 1:

How did you choose 50 mM NaCl without studying the other concentration range to all your experiments. Ionic strength of the solution is also may affect the electro-oxidation or reduction of carbon surfaces.

Response to Reviewer 1, Major Comment 1: 50 mM NaCl was employed in this work which is part of a larger work focused on corrosion in multi-material combinations comprised of metals and carbon-reinforced polymers. In corrosion studies, the chloride ion concentration is a factor in corrosion and its mechanisms. Previous work carried out in the laboratory informed the choice of 50 mM NaCl as it presents sufficiently aggressive conditions that enable corrosion of metallic material to occur and also permits mechanism(s) to be monitored using localized electrochemical techniques like scanning vibrating electrode technique (SVET), etc.  Much lower chloride concentration gives solutions with reduced electrical conductivity which could reduce signal strengths, while much higher concentrations increase the solution conductivity, noise, and the possibility of side reactions involving the chloride ions.

In addition, chloride evolution reactions are more likely at anodic potentials which were not used in the present work.

Reviewer 1: Major Comment 2:

Fig. 8 is meaning less, if you don’t show the FE-SEM images of the surface after oxidation.

It is required to show some spectrosocpic data (FT-IR) to confirm that surface has undergone oxidation or reduction.

Response to Reviewer 1, Major Comment 2: We strongly agree with your esteemed observation. Surface characterization of the carbon surfaces has now been carried out using scanning electron microscopy with energy dispersive x-ray spectroscopy (EDS) to monitor the surface oxygen and carbon compositions before and after polarization. The results are now presented in Table 1. Furthermore, a confocal  Raman study of the surfaces has now been carried out and the spectra are now presented in Figure 8 in the current version.

 Reviewer 1: Major Comment 3:

In this work, only electrochemical data are provided without any other supporting data from EDX, SEM, IR, contact-angle, etc.

Response to Reviewer 1, Major Comment 3: Surface characterization of the carbon surfaces has now been carried out using scanning electron microscopy with energy dispersive x-ray spectroscopy (EDS) to monitor the surface oxygen and carbon compositions before and after polarization. The results are now presented in Table 1. SEM images were not included as it does not furnish any new information to support our position. Instead, the EDS data which furnishes chemical compositional information of the respective surfaces is presented as Table 1.

Furthermore, a confocal  Raman study of the surfaces has now been carried out and the spectra are now presented in Figure 8 in the current version. 

Reviewer 1, Major Comment 4:

Introduction section should be updated and the following articles need to be cited, https://doi.org/10.1149/2754-2726/ac7c78

https://doi.org/10.3389/fmats.2022.906838

Response to Reviewer 1: Major Comment 4: The introduction section has now been updated with these and other references with the addition of more introductory materials on carbon surfaces and their evolutions.

Reviewer 1: Major Comment 5:

Minor English editing is required.

Response to Reviewer 1, Major Comment 5: The manuscript has now undergone some editing.

Reviewer 2 Report

Dear Authors

The manuscript is focused on the electrochemical response of CFRP, glassy carbon, and HOPG (Highly Ordered Pyrolytic Graphite) have been evaluated in quiescent 50 mM NaCl solution and their respective activities towards ORR ranked. Comparison of electrochemical data at potentials relevant to galvanic coupling to metals indicated that at these cathodic potential(s) CFRP surface is the most electrochemically active of the studied carbon surfaces.

The following suggestion and comments should be taken:

1. The overall English needs to be improved. Please seek guidance from a native English speaker if possible ("the" "a", commas, plural form and others could be corrected).

2. Line 5. Please delete dot after Media.

3. The authors could insert more numerical data into the Abstract for enhancement of the manuscript.

4. The introduction section needs enhancement 1-3 sentences about carbon fibres/carbon nanotubes composites and their different potential applications. Please cite: (1) Materials 2022, 15(12), 4252; https://doi.org/10.3390/ma15124252 (2) Materials 2021, 14(9), 2448; https://doi.org/10.3390/ma14092448 (3) Materials 2022, 15(12), 4270; https://doi.org/10.3390/ma15124270

5. Could the authors include the standard deviation of the used methods?

6. Figure 1. Please correct this image for better quality (the inscriptions).

7. Figure 5. Please put this image in shape a b (side by side).

8. Figure 6 the same as in Fig. 5.

9. Why authors used only ORR not other methods OER etc.? Please explain.

10. Authors are suggested to describe some future plans or potential applications of materials in conclusions.

Author Response

REVIEWER 2:

Dear Authors

he manuscript is focused on the electrochemical response of CFRP, glassy carbon, and HOPG (Highly Ordered Pyrolytic Graphite) have been evaluated in quiescent 50 mM NaCl solution and their respective activities towards ORR ranked. Comparison of electrochemical data at potentials relevant to galvanic coupling to metals indicated that at these cathodic potential(s) CFRP surface is the most electrochemically active of the studied carbon surfaces.

Reviewer 2, Major Comment 1:

The following suggestion and comments should be taken:

  1. The overall English needs to be improved. Please seek guidance from a native English speaker if possible ("the" "a", commas, plural form and others could be corrected).

Response Reviewer 2, Major Comment 1: The manuscript has now undergone some editing.

Reviewer 2, Major Comment 2:

  1. Line 5. Please delete dot after Media.

Response Reviewer 2, Major Comment 1: The dot after media has now been deleted.

Reviewer 2, Major Comment 3:

  1. The authors could insert more numerical data into the Abstract for enhancement of the manuscript.

Response Reviewer 2, Major Comment 3:  Thanks so much for this important observation. Numerical data on the average charges passed and capacitance have now been included in the abstract.

Reviewer 2, Major Comment 4:

  1. The introduction section needs enhancement 1-3 sentences about carbon fibres/carbon nanotubes composites and their different potential applications. Please cite: (1) Materials 2022, 15(12), 4252; https://doi.org/10.3390/ma15124252 (2) Materials 2021, 14(9), 2448; https://doi.org/10.3390/ma14092448 (3) Materials 2022, 15(12), 4270; https://doi.org/10.3390/ma15124270

Response Reviewer 2, Major Comment 4: The introduction section has now been updated with these and other references with the addition of more introductory materials on carbon surfaces and their evolutions.

Reviewer 2, Major Comment 5:

  1. Could the authors include the standard deviation of the used methods?

Response Reviewer 2, Major Comment 5: The errors in the plots have now been highlighted in Figures 6 and 7 to be fitting errors and standard errors from the mean values (SEM) respectively.

Reviewer 2, Major Comment 6:

  1. Figure 1. Please correct this image for better quality (the inscriptions).

Response Reviewer 2, Major Comment 6: Images of better quality have now been inserted as Fig. 1. In addition, the images have been re-arranged and the Figure title modified.

Reviewer 2, Major Comment 7:

  1. Figure 5. Please put this image in shape a b (side by side).

Response Reviewer 2, Major Comment 7: The images in Figure 5 have now been re-arranged as suggested.

Reviewer 2, Major Comment 8:

  1. Figure 6 the same as in Fig. 5.

Response Reviewer 2, Major Comment 8: The images in Figure 6 have now been re-arranged to appear side-by-side as suggested.

Reviewer 2, Major Comment 9:

  1. Why authors used only ORR not other methods OER etc.? Please explain.

Response Reviewer 2, Major Comment 9: Previous work in our laboratory indicates that in this system ORR is the predominant cathodic reaction at -1000 mV vs. SCE. The work is thus focused on ORR because this is the predominant reaction at the cathodic potential ranges CFRP is polarized to on galvanic coupling to most metals in multi-material assemblies now common in the aerospace and transport industries.  To mitigate galvanic corrosion of coupled metallic components in such hybrid structures, it is desirable to engineer coupling carbon surfaces to have lowered response to ORR. The present work demonstrates that among the 3 carbon surfaces investigated the CFRP surface manifests the greatest response to ORR. It is hoped that these results will motivate future efforts at engineering CFRP surfaces that have a reduced response to ORR similar to that observed on HOPG, as this would result in better corrosion-resistant hybrid structures.

On the other hand, OER is desirable for energy applications and research efforts are geared at enhancing the rates of OER on non-precious metal surfaces which includes carbon surfaces.

Reviewer 2, Major Comment 10:

  1. Authors are suggested to describe some future plans or potential applications of materials in conclusions.

Response Reviewer 2, Major Comment 10: This suggestion is highly appreciated.

The following sentences have now been added to the conclusion; “It is envisaged that these results will motivate future efforts at engineering CFRP surfaces that have a reduced response to ORR similar to that observed on HOPG, as this would result in better corrosion-resistant hybrid structures. In addition, the data from this work can be valuable as a baseline for ranking the performance of carbon surfaces engineered for reduced response to ORR. “

Round 2

Reviewer 1 Report

1. Author have revised the paper as per our suggestions. But, I do not know why they have cited enormous number of references in the revised article. Total reference in this article is 211, it is unusual for a research article. It has to be reduced, really all these references are essential. For a review paper, it may be ok.

2. There are some minor english editing is required.

-known differences in the electrochemical activity of basal 264 and edge carbon sites in carbon materials in corrosion mitigation in multi-material com- 265 binations.  so many 'in'...... like this there should be some english language editing is required throughout the manuscript. 

Author Response

Reviewer 1

Reviewer 1, Comment 1:

  1. Author have revised the paper as per our suggestions. But, I do not know why they have cited enormous number of references in the revised article. Total reference in this article is 211, it is unusual for a research article. It has to be reduced, really all these references are essential. For a review paper, it may be ok.

Response to Reviewer 1, Comment 1:

First, We wish to convey our immense gratitude for your esteemed time, meticulous critique, and suggestions that has resulted in improvements in the manuscript.

The references have now been pruned down ( to the most relevant) in the present version.

We concede that the total references cited in the last version increased from 180 to 211 and that this is similar to the reference count in review articles.

This was due to both reviewers’ comments in the first review round which required us to enhance the introduction and employ additional surface analysis techniques (Raman spectroscopy and EDS analysis) from which we had to draw conclusion to argue our deductions.

In addition, 3 materials were studied with a focus on one process (ORR), using a variety of tests methods. As there had been significant number of studies on these materials and process, it was necessary to properly set the present study in perspective in the light of earlier works.

Reviewer 1, Comment 2:

  1. There are some minor english editing is required.

-known differences in the electrochemical activity of basal 264 and edge carbon sites in carbon materials in corrosion mitigation in multi-material com- 265 binations.  so many 'in'...... like this there should be some english language editing is required throughout the manuscript.

Response to Reviewer 1, Comment 2:

The entire manuscript has now undergone English editing.

The issues raised have been addressed and the fluency of the manuscript improved.  

Changes now effected in the current version can be observed in the current version of the manuscript with tracked changes.

Reviewer 2 Report

The authors have addressed all comments and the manuscript can be published as is.

Author Response

Reviewer 2

Reviewer 2, Comment 1:

The authors have addressed all comments and the manuscript can be published as is.

Response to Reviewer 1, Comment 1: Thanks so much for your esteemed time and the earlier suggestions that has resulted in improvements in the manuscript.